# House Dust Avoidance during Pregnancy and Subsequent Infant Development: The Japan Environment and Children’s Study

**DOI:** 10.3390/ijerph18084277

**Published:** 2021-04-17

**Authors:** Kenta Matsumura, Kei Hamazaki, Akiko Tsuchida, Hidekuni Inadera

**Affiliations:** 1Toyama Regional Center for Japan Environment and Children’s Study, University of Toyama, 2630 Sugitani, Toyama 930-0194, Japan; keihama@med.u-toyama.ac.jp (K.H.); aktsuchi@med.u-toyama.ac.jp (A.T.); inadera@med.u-toyama.ac.jp (H.I.); 2Department of Public Health, Faculty of Medicine, University of Toyama, 2630 Sugitani, Toyama 930-0194, Japan

**Keywords:** housekeeping, dust, infant, diagnostic screening programs, psychomotor disorders, communication disorders

## Abstract

House dust, well known for causing allergy, contains chemicals that are harmful to fetal neurodevelopment. However, whether countermeasures for house dust allergy, such as frequent use of vacuum cleaners, frequent airing of futons, and the usage of anti-mite covers during pregnancy, are related to subsequent reduced risk of infant developmental delay remains unknown. Therefore, we examined this association by analyzing 81,106 mother-infant pairs who participated in a nationwide birth cohort in Japan. Infant developmental delays at 6 and 12 months postpartum were assessed using the Ages and Stages Questionnaire, Third Edition. A generalized linear model analysis was used to derive adjusted odds ratios (AORs) with the lowest care frequency as reference, while controlling 22 covariates. Our analysis showed that the above-mentioned cleaning measures were overall associated with a reduced risk of developmental delays, both at 6 and 12 months postpartum (AOR varied from 0.73 to 0.95, median: 0.84). Additionally, risks tended to decrease with an increase in the cleaning frequency. In conclusion, a negative dose-response association existed between these measures during pregnancy and infant developmental delay. Our results identify a potential role of frequent vacuum cleaning, airing bedding, and usage of anti-mite bedding covers in promoting intact infant development.

## 1. Introduction

House dust has been identified as one of the major causes of allergy. It contains several allergens such as pollen, mites, human and pet skin that mites feed on, and pet hair [1]. Existing countermeasures for house dust allergy, such as asthma and atopic dermatitis, include the frequent usage of a vacuum cleaner, dry-steam cleaning, air purifiers [2], and covering of mattresses and pillows [3], although these anti-house dust treatments are not always effective [4].

Besides allergens, house dust also contains various chemicals such as lead [5], di(2-ethylhexyl) phthalate (phthalate ester) [6], polycyclic aromatic hydrocarbons (PAHs) [7], and polybrominated diphenyl ethers [8]. These chemicals are known to interfere with normal infant neurodevelopment [9,10,11]. For example, fetal exposure to PAHs, measured as mother’s exposure to PAHs, is negatively associated with the development of brain white matter, cognition, and behavior in later childhood [12]. Since a series of findings have indicated that house dust is one of the principal sources of exposure to these neurotoxicants [5,7,8,13], the above-mentioned countermeasures for allergy may also be negatively associated with the occurrence of infant neurodevelopmental delays. However, the relationship between such countermeasures and infant developmental delay remains unknown.

Therefore, this study examined the prospective relationship between home cleaning measure for house dust avoidance during pregnancy and subsequent development, with a specific focus on frequency of vacuum cleaning, airing of bedding, and anti-mite bedding cover use. We assessed infant development until 1 year of age using a large dataset from the Japan Environment and Children’s Study (JECS), a nationwide birth cohort with >80,000 records while controlling for possible confounders such as basic characteristics and dwelling environment. We hypothesize that frequent floor and bedding cleaning activities are related to a reduced risk of developmental delay in infants.

## 2. Materials and Methods

### 2.1. Study Design and Sample

The JECS is a nationwide, government-funded birth cohort study investigating the impact of various environmental factors on children’s health and development. In the JECS, 103,062 pregnancies were registered between January 2011 and March 2014 via face-to-face recruitment at public agencies across 15 regions, including both rural and urban locations throughout Japan (from the subpolar northern island, Hokkaido, to the southern island, Okinawa). The sample size was determined in advance to maintain adequate statistical power enough to detect rare diseases or conditions such as with ≤1% prevalence. The eligibility criteria for participants (expecting mothers) were as follows: (1) Residence in the study areas at the time of recruitment and expected to reside continuously in Japan for the foreseeable future; (2) expected delivery date between 1 August 2011 and mid-2014; (3) ability to comprehend the Japanese language and complete the self-administered questionnaire. Those residing outside the study areas, even if they visited the cooperating health care providers within the study areas, were excluded from the study. Follow-ups were primarily conducted via scheduled in-hospital check-ups until 1 month postpartum and via mailed letters at 6 and 12 months postpartum. Data were acquired using self-administered questionnaires or transcriptions of medical records recorded by physicians, midwives, nurses, and research coordinators.

The dataset used in this study, jecs-an-20180131, was released in March and updated in December of 2018, and included measurements for the first and second/third trimester, as well as for 0, 1, 6 and 12 months postpartum. Among the 103,062 participants in this dataset, 5647 were excluded due to multiple registrations, 949 due to multiple births, and 3676 following miscarriages or stillbirths. Among the remaining 92,790 pregnancies with singleton live births, 3281 were excluded due to missing information on exposure variables during pregnancy. Another 8403 pregnancies were further excluded due to a lack of responses to the posted Ages and Stages Questionnaire^®^, Third Edition (ASQ-3) [14], both at 6 and 12 months postpartum. Thus, data from 81,106 mother-child pairs were analyzed (Figure 1).

### 2.2. Measures

#### 2.2.1. Exposure

A total of four variables measured during the second or third trimester were used as exposure variables: (1) Frequency of cleaning the living room floor with a vacuum cleaner as an average throughout the year (daily, a few times a week, once a week, 1–2 times a month, a few times a year, almost never, or never); (2) frequency of cleaning the futon (Japanese mattress and blanket for bedding) with a vacuum cleaner as an average throughout the year (a few times a week, once a week, 1–2 times a month, a few times a year, almost never, or never); (3) frequency of airing the futon as an average throughout the year (a few times a week, once a week, 1–2 times a month, a few times a year, almost never, or never); (4) usage of anti-mite covers for the futon or bedding after conceiving (yes or no).

Although the average frequency of cleaning the bedroom floor with a vacuum cleaner throughout the year was also measured, this variable was highly associated with the frequency of cleaning the living room floor (Spearman’s ρ (rho) = 0.79) and was therefore not included in our analyses.

All variables (except for usage of anti-mite covers) were recategorized so that each variable consisted of four categories.

#### 2.2.2. Outcomes

Children’s development at 6 and 12 months after birth was assessed using the ASQ-3, an age-specific, structured, parent-completed child monitoring system [14]. The ASQ-3 is a set of well-validated, globally used questionnaires that is recommended by the United Nations Children’s Fund to identify potential delays and help determine which children need further assessment or ongoing monitoring [15]. The Japanese version of the ASQ-3 has also been validated [16] and has already been used in previous studies [17,18,19]. The ASQ-3 assesses the following five areas of development: (1) Communication: Language skills, such as babbling, vocalizing, listening, and understanding; (2) gross motor: Arm, body, and leg movements to move and play; (3) fine motor: Hand and finger movements; (4) problem solving: Problem-solving skills, learning, and playing with toys; (5) personal-social: Self-help skills, solitary social play, and play with toys and others. Each area consists of six developmental items. The parents’ responses (yes, sometimes, and not yet) were counted as 10, 5 and 0 points, respectively, and the total score for each area ranged from 0 to 60 points. The screen-positive cases for each area were defined as those with scores on or below the respective threshold values, which were set at −2 standard deviations from the mean (z-score ≤ −2) to yield a sensitivity of 85–92%, a specificity of 78–92%, and a positive predictive value of 32–64% [14]. Taking early delivery into account, if the parent’s completion date at 6 and 12 months was not within ±1 month from the estimated delivery date, the data were treated as missing values in accordance with the scoring guidelines.

#### 2.2.3. Covariates

Unless otherwise specified, covariates were measured during the second or third trimester. We used the following variables as covariates for mothers: maternal age (<25, 25–<30, 30–<35 or ≥35 years), pre-pregnancy body mass index asked during the first trimester (<18.5, 18.5–<25 or ≥25 kg/m^2^), parity (primipara or multipara), history of allergy measured during the first trimester (yes or no), Kessler Psychological Distress Scale (K6) score (<5, 5–12, ≥13) [20,21,22], smoking status (never, former, or current), alcohol intake (never, former, or current), number of hours spent outdoors (<1, 1–<2, 2–<3 or ≥3 h), physical activity corresponding to over 10 min of walking a day (yes or no), quintile of folic acid intake measured using food frequency questionnaire (≤153, 154–203, 204–257, 258–337 or ≥338 µg) [23,24], marital status assessed during the first trimester (married, single, divorced, or widowed), highest educational level (≤12, >12–<16 or ≥16 years), employment status (yes or no), and annual household income (<4, 4–<6 or ≥6 million Japanese Yen). As covariates for dwelling environment, we used type of residence (wooden detached house, steel-frame detached house, wooden multiple dwelling house/apartment, steel-frame multiple dwelling house/apartment, or other), number of rooms in the house or apartment (≤2, 3, 4, 5 or ≥6), living room flooring material (*tatami* [Japanese straw floor covering], carpet on tatami, wooden flooring/tiles, carpet on wooden flooring/tiles, or other), having a pet (yes or no) [18], usage of air purifiers (yes or no) [17], age of house or apartment building (<1, 1–<3, 3–<5, 5–<10, 10–<20 or ≥20 years, or unknown), house renovation or interior completion after conception (yes or no), and number of years living in the current place of residence (<1, 1–<3, 3–<5, 5–<10, 10–<20 or ≥20 years). These covariates comprise the general and socio-economic status and dwelling environment as well as variables that have potential impact on exposure and outcome. Potential mediators were not used as covariates. All continuous variables were categorized in advance in the event that complex relationships should exist. The categorization of these variables was conducted according to usual medical practice or common practice in Japan [25,26] and is shown in Table 1.

### 2.3. Statistical Analysis

Background differences according to the exposure variables were examined using a chi-squared (χ^2^) test. Given our large sample size and that minor differences could be easily detected, we relied not only on the *p*-value but also on the effect size Cramer’s V, the values being independent of the sampling size [27].

Generalized linear models, with logit set as a link function, were used to calculate crude and adjusted odds ratios (ORs) and their 95% confidence intervals (CIs). Exposure variables were: (1) Frequency of cleaning the living room floor with a vacuum cleaner, (2) frequency of cleaning the futon, (3) frequency of airing the futon, and (4) anti-mite cover use. The lowest category in each variable and non-use of anti-mite covers were set as the reference. Outcome variables were the total number of screen-positive cases (ranging from 0 to 5) each at 6 and 12 months postpartum. For example, if an infant falls in cases of communication, fine motor, and problem solving, the total number equals 3. In addition, ORs and CIs of screen-positive cases in each developmental area; (a) communication, (b) gross motor, (c) fine motor, (d) problem solving, (e) personal-social (in ASQ-3), were also calculated. The forced entry method was used to include covariates in the multivariable analysis.

SAS 9.4 (SAS Institute Inc., Cary, NC, USA) was used for all statistical analyses.

### 2.4. Missing Data

Of the 81,106 mother-infant pairs included in the study, the missing-data rate was ≤1% for all covariates except for parity (2.51%; *n* = 2032), number of years living in the current place of residence (2.90%; *n* = 2355), physical activity (3.12%; *n* = 2530), average number of hours spent outdoors (3.78%; *n* = 3046), and annual household income (6.60%; *n* = 5352). Each score of the five areas of the ASQ-3 had a <5.95% (max *n* = 4820) missing-data rate at 6 months and <10.51% (max *n* = 8518) at 12 months.

Imputation was conducted using chained equations [28] to obtain 10 imputed data sets. All data were imputed simultaneously, regardless of the measured time points. When conducting multiple imputations, auxiliary variables that were likely associated with the analyzed variables were also included so as not to violate the assumption of missing at random. The estimates from each dataset after multiple imputations were combined using the Rubin’s rules [29].

## 3. Results

### 3.1. Basic Characteristics

Data from 81,106 mother-child pairs were analyzed. The mothers’ mean age during pregnancy was 31.1 ± 4.99 (SD) years, mean body mass index before pregnancy was 21.2 ± 3.24 (SD), 43.3% of the mothers were primipara, 35.1% had less than 12 years of education, 39.5% had an annual household income of less than four million Japanese yen, 4.0% were current smokers, and 2.7% were current alcohol drinkers during pregnancy. The basic characteristics of mothers and dwelling environments during pregnancy are presented in Table 1.

The top three covariates related to each exposure were as follows: not employed (Cramer’s V = 0.297), multipara (V = 0.283), and longer hours spent outdoors (V = 0.085) for vacuum cleaning of the living room (Appendix A); usage of air purifiers (V = 0.082), multipara (V = 0.071), and lower education level (V = 0.066) for vacuum cleaning of the futon (Appendix A); multipara (V = 0.153), not employed (V = 0.128) and longer hours spent outdoors (V = 0.086) for airing the futon (Appendix A); usage of air purifiers (V = 0.073), folic acid intake (V = 0.036), and lower age of house or apartment building (V = 0.029) for usage of anti-mite covers (Appendix A S4).

The frequency of each exposure measure is listed in Table 2.

### 3.2. Main Analyses

All generalized variance inflation factors were below 2.45, and therefore, no multicollinearity was detected among any of the covariates.

The mean and standard deviation of the total numbers of screen-positive cases over the five developmental areas were 0.116 ± 0.106 (SD) at 6 months and 0.222 ± 0.273 (SD) at 12 months. The numbers, crude, and adjusted ORs, according to the exposure variables, are shown in Figure 2. Adjusted ORs varied from 0.74–0.93 (median value = 0.84) at 6 months and 0.73–0.95 (median value = 0.84) at 12 months. Trend test for adjusted models revealed that all *p*-values for trend were <0.001.

The prevalence, as well as crude and adjusted ORs for developmental delay in each developmental area at 6 and 12 months postpartum, according to the exposure variables, are presented in Appendix A. Adjusted ORs varied from 0.69–1.04 (median value = 0.83) at 6 months and 0.66–1.01 (median value = 0.86) at 12 months.

## 4. Discussion

To our knowledge, this is the first study to examine the prospective association of home cleaning measures for house dust avoidance during pregnancy, including frequent vacuum cleaning, airing bedding, and usage of anti-mite bedding covers, with subsequent infant development until 1 year of age. To this end, we used a nationwide birth cohort study dataset (*n* > 80,000) while controlling for up to 22 potential confounders with a low level of multicollinearity. Our analyses revealed that adjusted ORs of these cleaning measures for house dust removal, the frequent use of vacuum cleaners, frequent airing of futons, and the usage of anti-mite covers during pregnancy, in screen-positive cases of developmental delay, as per the ASQ-3, were significantly lower compared to the references (which were set to the lowest frequency). This finding remained significant across five areas both at 6 and 12 months postpartum. In addition, the adjusted ORs tended to decrease with an increase in cleaning frequency. Therefore, we found a negative dose-response relationship between house dust avoidance during pregnancy and infant developmental delay.

Fetal exposure to PAHs that are typically found in house dust is associated with decreased birth weight and smaller head circumference [30] as well as reduced brain white matter, cognition, behavior at 8 years [12], and intelligence quotient (IQ) at 5 years of age [9]. Although the detailed mechanisms underlying the relationship between high fetal exposure to PAHs and impaired child neurodevelopment remain unknown, several possible pathways, such as binding to receptors prerequisite for intact fetal growth [9,30] and inflammation and oxidative stress [31], may be involved. Although we observed a negative association between the frequent use of vacuum cleaners, frequent airing of the futon, as well as usage of anti-mite covers and developmental delay until 1 year of age, it remains unclear whether such interventions relate to brain white matter, cognition, behavior, and IQ at later ages. It will be of great importance to examine these aspects as well as the occurrence of developmental delay measured by ASQ-3 at later ages.

It should be noted that conclusions regarding causality cannot be drawn from our findings. Consequently, the observed relationships do not necessarily guarantee that increasing the frequency of cleaning or airing, or introducing anti-mite covers will always prevent infants from developmental delay. In fact, the factors associated with reduced risks for developmental delay through these interventions remain unknown. Thus, further studies conducting randomized controlled trials to address causality, and in-depth studies, such as elucidating factors associated with reduced risk for developmental delay in relation to these interventions and sampling the infant and mother’s blood for known household neurotoxins, are warranted.

Frequent cleaning of the floor with a vacuum cleaner and airing of the futon were strongly associated with multipara and unemployment. Mothers with young children are typically required to clean frequently, and being unemployed is virtually equal to being a full-time homemaker in Japan, this allows for daily cleaning and airing of futons. Since working mothers are less frequently at home compared to unemployed mothers, this may reduce the time that can be dedicated to cleaning. Interestingly, a history of maternal allergy was not strongly related to the frequency of these cleaning behaviors, even anti-mite cover use. Thus, the motivation for these living environment cleaning behaviors seems to be based on the natural necessity and capacity in daily life, though the cleaning tasks do not have to be performed by the mother, but by any domestic partner/husband, care taker, home helper, or older child.

The study has two important strengths. First, we included data from an ongoing birth cohort with a large sample of ≥80,000 recruited on a nationwide scale between 2011 and 2014. Therefore, the sample could be regarded as representative of recent Japanese mothers. Second, although we followed expectant mothers from early pregnancy to 1 year after childbirth, a relatively high response rate was maintained. Consequently, selection bias caused by loss to follow-up was probably small.

The are several limitations to this study. First, we asked only about the frequency of cleaning using vacuum cleaners or airing and did not include other means of removing house dust, such as sweeping or wiping the floor with a broom or a dust cloth. Thus, answering “almost never or never” to the question on the frequency of cleaning with a vacuum cleaner did not necessarily indicate a higher level of house dust. Second, we did not ask about manufacturer names, even though it is known that different vacuum cleaners vary in their cleaning performance, and this may have affected outcomes [32]. Third, we did not have objective data for the decrease in indoor house dust levels after cleaning. While some studies reported that cleaning with vacuum cleaners does result in house dust removal [33,34], others obtained contrasting results [4]. Fourth, although the ASQ-3 is a validated parent-completed questionnaire, its aim is screening, not diagnosis. Thus, the reduced accuracy in assessing signs of developmental delay in this study, as compared with a diagnosis by a professional, should be acknowledged. Finally, while our sample consisted of the JECS nationwide birth cohort, the extent to which our results can be generalized to other living environments such as bed, sleeping mat, and bed-bug protector, cannot be determined. Nevertheless, given that the underlying mechanism of observed reduced risk is actually involved in the removal of many chemicals in house dust, the finding has the potential to be applicable to a wide population.

## 5. Conclusions

Our results derived from 81,106 mother-child pairs indicate a potential role of house dust avoidance, such as frequent vacuum cleaner use, frequent futon airing, and use of anti-mite covers, in promoting intact infant development. This study provides a rationale for conducting randomized controlled trials to address the effect of these measures on inhibiting developmental delay, as well as for in-depth studies elucidating the factors associated with a reduced risk of developmental delay in relation to these home cleaning measures.

## Figures and Tables

**Figure 1 ijerph-18-04277-f001:**
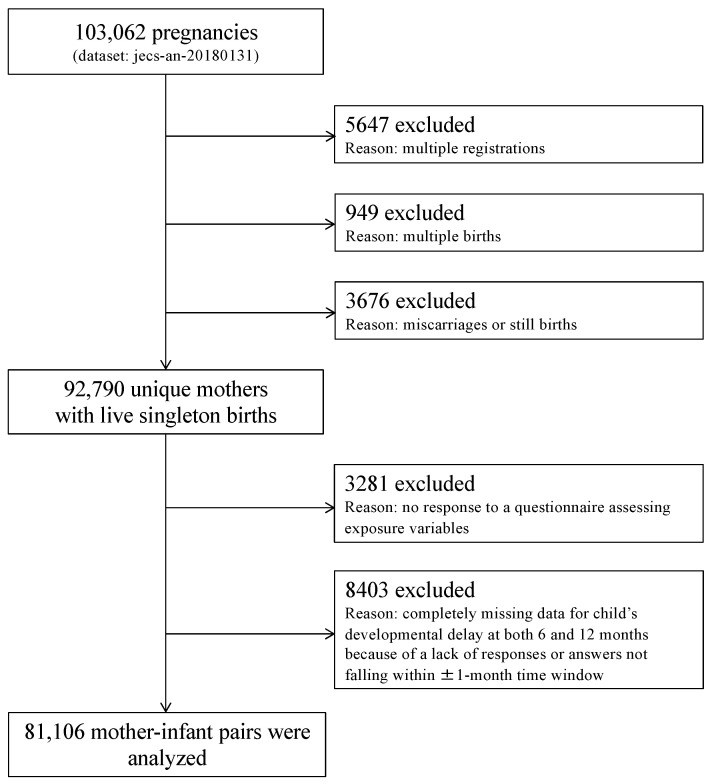
Study flow chart.

**Figure 2 ijerph-18-04277-f002:**
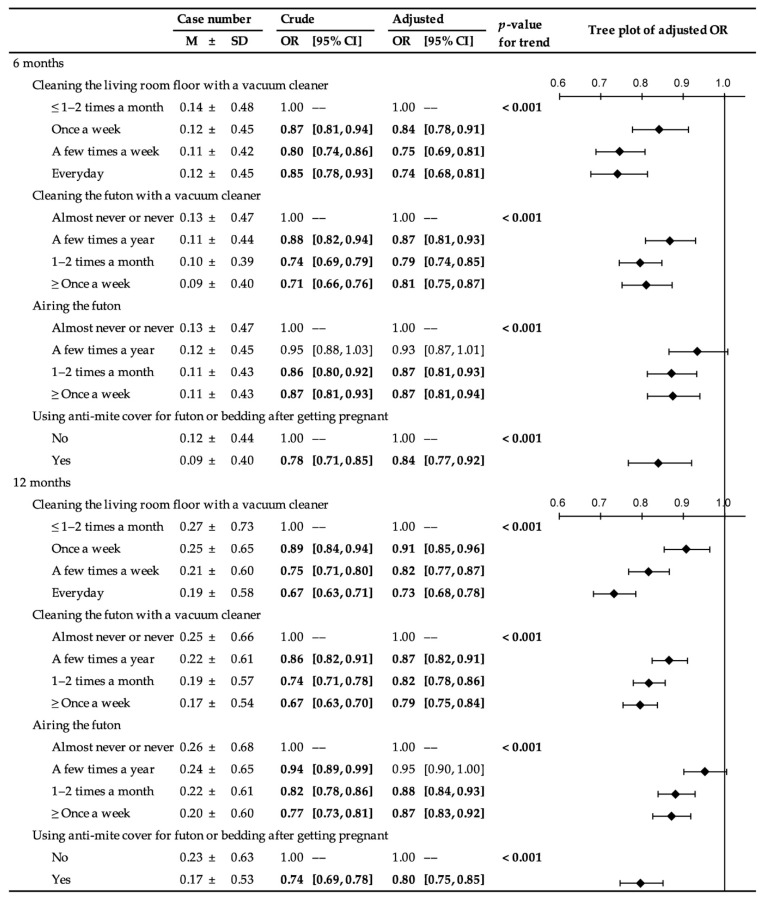
Total number of screen-positive cases over the five developmental areas in the ASQ-3 (case number) and their odds ratios according to the exposure variables (based on imputed data for the 81,106 infants in this study). Boldface indicates statistical significance at the level of 5%. OR, odds ratio; CI, confidence interval; ASQ-3, Ages and Stages Questionnaire, Third Edition. Crude = crude model. Adjusted = model adjusted for maternal age, body mass index, parity, history of allergy, psychological distress, smoking status, alcohol intake, number of hours spent outdoors, physical activity, folic acid intake, marital status, highest educational level, employment status, annual household income, type of residence, number of rooms in the house or apartment, living room flooring material, having a pet, usage of air purifiers, age of house or apartment building, house renovation or interior completion after becoming pregnant, and number of years living in the current place of residence. *p*-values for trend were calculated for adjusted models.

**Table 1 ijerph-18-04277-t001:** Characteristics of mothers and dwelling environments.

Variable	Category	*n*	(%)
Age, y	<25	7971	(9.8)
	25–<30	23,069	(28.4)
	30–<35	28,887	(35.6)
	≥35	21,171	(26.1)
Pre-pregnancy body mass index ^a^, kg/m^2^	<18.5	13,129	(16.2)
	18.5–<25	59,722	(73.7)
	≥25	8212	(10.1)
Parity	Primipara	34,250	(43.3)
	Multipara	44,824	(56.7)
History of allergy ^a^	No	40,225	(49.8)
	Yes	40,548	(50.2)
Kessler Psychological Distress Scale (K6)	<5	58,089	(71.7)
	5–12	20,482	(25.3)
	≥13	2449	(3.0)
Smoking status	Never	47,394	(58.9)
	Former	29,877	(37.1)
	Current	3245	(4.0)
Alcohol intake	Never	26,915	(33.4)
	Former	51,387	(63.8)
	Current	2212	(2.7)
Number of hours spent outdoors, hour	<1	14,905	(19.1)
	1–<2	37,511	(48.1)
	2–<3	12,306	(15.8)
	≥3	13,338	(17.1)
Physical activity	No	18,041	(23.0)
	Yes	60,535	(77.0)
Quintile of folic acid intake, μg	≤153	15,764	(19.4)
	154–203	16,285	(20.1)
	204–257	16,400	(20.2)
	258–337	16,425	(20.3)
	≥338	16,224	(20.0)
Marital status ^a^	Married	77,024	(95.7)
	Single	2803	(3.5)
	Divorced or widowed	628	(0.8)
Highest education level, y	≤12	28,351	(35.1)
	>12–<16	34,413	(42.6)
	≥16	18,067	(22.4)
Employed	No	36,658	(45.5)
	Yes	43,878	(54.5)
Annual household income, million yen	<4	29,903	(39.5)
	4–<6	25,223	(33.3)
	≥6	20,628	(27.2)
Type of residence	Wooden detached house	33,280	(41.2)
	Steel-frame detached house	5108	(6.3)
	Wooden multiple dwelling house/apartment	9944	(12.3)
	Steel-frame multiple dwelling house/apartment	31,605	(39.2)
	Other	771	(1.0)
Number of rooms in the house/apartment	≤2	15,212	(18.8)
	3	26,151	(32.4)
	4	15,389	(19.1)
	5	12,093	(15.0)
	≥6	11,883	(14.7)
Living room flooring materials	Tatami (Japanese straw floor covering)	9161	(11.3)
	Carpet on tatami	7186	(8.9)
	Wooden flooring/tiles	28,703	(35.5)
	Carpet on wooden flooring/tiles	34,390	(42.5)
	Other	1467	(1.8)
Having a pet	No	62,503	(77.2)
	Yes	18,451	(22.8)
Usage of air purifiers	No	39,830	(49.2)
	Yes	41,152	(50.8)
Age of house/apartment building, y	<1	4593	(5.7)
	1–<3	9138	(11.3)
	3–<5	7701	(9.5)
	5–<10	12,519	(15.5)
	10–<20	18,829	(23.3)
	≥20	20,253	(25.1)
	Unknown	7637	(9.5)
House renovation/interior finishing after getting pregnant	No	78,141	(96.8)
	Yes	2560	(3.2)
Number of years living in the current place of residence	<1	5373	(6.8)
	1–<3	34,072	(43.3)
	3–<5	17,840	(22.7)
	5–<10	14,018	(17.8)
	10–<20	4202	(5.3)
	≥20	3246	(4.1)

Note: Unless otherwise specified, covariates were measured during the second or third trimester. ^a^ measured or asked during the first trimester.

**Table 2 ijerph-18-04277-t002:** Exposure variables with respective correlation matrices.

	*n*	(%)	Matrix of Spearman’s ρ (rho)			
			A	B	C	D
A. Frequency of cleaning the living room floor with a vacuum cleaner			––	0.18 ***	0.25 ***	−0.04 ***
≤1–2 times a month	6795	(8.4)				
Once a week	25,367	(31.3)				
A few times a week	35,362	(43.6)				
Everyday	13,582	(16.7)				
B. Frequency of cleaning the futon with a vacuum cleaner				––	0.10 ***	−0.10 ***
Almost never or never	46,045	(56.8)				
A few times a year	10,286	(12.7)				
1–2 times a month	13,265	(16.4)				
≥Once a week	11,510	(14.2)				
C. Frequency of airing the “Futon”					––	0.01 *
Almost never or never	9591	(11.8)				
A few times a year	15,673	(19.3)				
1–2 times a month	30,118	(37.1)				
≥Once a week	25,724	(31.7)				
D. Using anti-mite cover for futon or bedding after getting pregnant						––
No	74,757	(92.2)				
Yes	6349	(7.8)				

Note: All were measured during the second or third trimester. *** *p* < 0.001, * *p* < 0.05.

## Data Availability

The data used to derive our conclusions are unsuitable for public deposition owing to ethical restrictions and the specific legal framework in Japan. Specifically, it is prohibited by the Act on the Protection of Personal Information (Act No. 57 of 30 May 2003, amended 9 September 2015) to publicly deposit data containing personal information. The Ethical Guidelines for Epidemiological Research enforced by the Japan Ministry of Education, Culture, Sports, Science and Technology and the Ministry of Health, Labour and Welfare also restrict the open sharing of epidemiologic data. All inquiries regarding access to the data should be sent to jecs-en@nies.go.jp. The person responsible for handling inquiries sent to this e-mail address is Dr. Shoji F. Nakayama, JECS Program Office, National Institute for Environmental Studies.

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
