# Peer review of "House Dust Avoidance during Pregnancy and Subsequent Infant Development: The Japan Environment and Children’s Study"

_ijerph, 2021, doi:10.3390/ijerph18084277_

Round 1
Reviewer 1 Report
Due March 31, 2021
Review request: International Journal of Environmental Research and Public Health
Type of manuscript: Article
Manuscript ID: ijerph-1157034
Title: Hour dust avoidance during pregnancy and subsequent infant neurodevelopment: the Japan environment and children’s study
Synopsis
The Japan Environment and Children’s Study (JECS) containing 103,062 pregnancy datum was selected for data on mother-infant (6th and 12th month time point) pairs and four exposure variables (1. vacuum cleaning of living area, 2. cleaning of sleeping mat, 3. airing of sleeping mat, and 4. dust-mite protector), resulting in a dataset of 81,106 mother-infant pairs (Figure 1). Outcome variables were the total number of positive cases (developmental deficiencies identified by two standard deviations or more in a Z-distribution) by a parent-completed child monitoring system, the Ages & Stages Questionnaires third edition (ASQ-3TM) for each of five assessment areas (communication, gross motor, fine motor, problem solving, and personal-social). Characteristics of mothers and dwelling environments were listed in Table 1. Figure 2 summarizes the association between the higher exposure (cleaning) to the lower outcome (lower developmental scores below the cutoff value). A dose-dependent reduction of the outcome to the exposure was also observed. Covariates for each exposure factor are summarized with effect size in the supplemental tables 1 through 4. Finally the adjusted odds ratio after covariates were controlled is indicated with a 95% Confidence Interval in supplemental Table S5.
Reviewer's conflict of interest: None
General comment to authors
Thank you very much for letting me review this well-composed manuscript. Although the chemicals and a heavy metal mentioned in line 40 – 44 are known biological neurotoxins and house dust is associated with these chemicals, the outcome instrument (a screening tool for developmental landmarks) is not directly linked to neurodevelopmental deficiencies. They are simply the deficiencies in behavioral landmarks at a chronological time point. I searched references #14 and #15 with the term “neurodevelopment,” but was unable to find that these behavioral landmarks imply the development of the nervous system. For example, as mentioned in the Discussion section, a stay-at-home mother can spend more time with a child or infant to coach and encourage these behaviors relative to employed mothers. Can a covariate, hours spent with the infant, be controlled? Did the blood from the infants who were identified as cases contain the aforementioned neurotoxins? Additional data may be required in order to appropriately use the term “neurodevelopment.”
Specific comments are listed below:
- Title: Please re-evaluate the use of “neurodevelopment” or state the link between the ASQ-3 and deficient neuronal development (more than behavioral landmarks).
- Abstract: Well written. The discussion on harmful neurotoxins is fine. It is also fine to describe the attempts to capture neurodevelopmental delay as parent-monitored behavioral landmarks.
- Introduction: Line 45 – 50 is the gap in knowledge and Line 51 – 58 describes a priori research-designs. Both are well-written.
- Methods:
- Line 61 – 66: A well-described parent dataset from JECS.
- Line 66 – 67: N=80,000 was justified by estimation of <1% prevalence value. In fact, the cut off value was two standard deviations using a one sided Z distribution. Thus, the predetermined prevalence was about 2.3%, which was less than 2,000 cases per exposure category.
- Line 100 – 103: The “cleaning the bedroom floor with a vacuum cleaner” could be excluded using collinear analysis. The logic of running Spearman or Pearson to combine variables was understood.
- Line 110 – 116: “Culturally and developmentally appropriate stimulated activity” (Ref 15, p45) can be emphasized rather than arguing for “normal neurological development” (Line 114).
- Line 125-126: Psychometrics are well-described.
- Line 131 – 157: A thorough explanation of covariates was appreciated.
- Line 160: Spell out “chi-square” before the Greek letter.
- Results:
- Line 193: Please add an explanation that “an annual income” is for the household, per person, or for the mother.
- As for indentation, line 227 – 234 is a part of the caption for Figure 2.
- Two tables and one figure are adequate for the results. The supplemental tables are also well presented. The tree plot is very nice to have.
- Discussion:
- Line 259 – 265: the conservative cautionary summary statements were well-stated.
- Line 270 – 273: “In-depth studies” include, but are not limited to, sampling the infant and mother’s blood for known household neurotoxins.
- Line 275 – 273: To clarify, the cleaning tasks do not have to be performed by the mother, but by any domestic partner/husband, care taker, home helper or older child.
- Line 303: “Other population” can refer to other cultural norms and languages used to describe the living environment. “Futon” can be a bed or sleeping mat, and “anti-mite covers” can be a bed-bug protector.
- Conclusion: No comment
End of review.
Author Response
Response to Reviewer 1 Comments
Synopsis
The Japan Environment and Children’s Study (JECS) containing 103,062 pregnancy datum was selected for data on mother-infant (6th and 12th month time point) pairs and four exposure variables (1. vacuum cleaning of living area, 2. cleaning of sleeping mat, 3. airing of sleeping mat, and 4. dust-mite protector), resulting in a dataset of 81,106 mother-infant pairs (Figure 1). Outcome variables were the total number of positive cases (developmental deficiencies identified by two standard deviations or more in a Z-distribution) by a parent-completed child monitoring system, the Ages & Stages Questionnaires third edition (ASQ-3TM) for each of five assessment areas (communication, gross motor, fine motor, problem solving, and personal-social). Characteristics of mothers and dwelling environments were listed in Table 1. Figure 2 summarizes the association between the higher exposure (cleaning) to the lower outcome (lower developmental scores below the cutoff value). A dose-dependent reduction of the outcome to the exposure was also observed. Covariates for each exposure factor are summarized with effect size in the supplemental tables 1 through 4. Finally the adjusted odds ratio after covariates were controlled is indicated with a 95% Confidence Interval in supplemental Table S5.
Reviewer's conflict of interest: None
General comment to authors
Thank you very much for letting me review this well-composed manuscript. Although the chemicals and a heavy metal mentioned in line 40 – 44 are known biological neurotoxins and house dust is associated with these chemicals, the outcome instrument (a screening tool for developmental landmarks) is not directly linked to neurodevelopmental deficiencies. They are simply the deficiencies in behavioral landmarks at a chronological time point. I searched references #14 and #15 with the term “neurodevelopment,” but was unable to find that these behavioral landmarks imply the development of the nervous system. For example, as mentioned in the Discussion section, a stay-at-home mother can spend more time with a child or infant to coach and encourage these behaviors relative to employed mothers. Can a covariate, hours spent with the infant, be controlled? Did the blood from the infants who were identified as cases contain the aforementioned neurotoxins? Additional data may be required in order to appropriately use the term “neurodevelopment.”
Response: Thank you very much for your evaluation. Based on your comment, we understand the inappropriateness of using the term “neurodevelopment”. In fact, we were unable to analyse the time mothers spent with their children nor the neurotoxins in their blood. Thus, we have removed the term “neuro” from the term “neurodevelopment” in the revised manuscript, except for the place actually referring to neurodevelopment, such as gray matter decreases.
Specific comments are listed below:
- Title: Please re-evaluate the use of “neurodevelopment” or state the link between the ASQ-3 and deficient neuronal development (more than behavioral landmarks).
Response: Following your suggestion, we have revised the title as follows: “House dust avoidance during pregnancy and subsequent infant development: The Japan Environment and Children’s Study.” - Abstract: Well written. The discussion on harmful neurotoxins is fine. It is also fine to describe the attempts to capture neurodevelopmental delay as parent-monitored behavioral landmarks.
Response: Thank you for this comment. Since we have also replaced “neurodevelopment” with “development” at instances it was iappropriately used in the abstract;it now seems unnecessary to describe “the attempts to capture neurodevelopmental delay as parent-monitored behavioral landmarks.” - Introduction: Line 45 – 50 is the gap in knowledge and Line 51 – 58 describes a priori research-designs. Both are well-written.
- Methods:
- Line 61 – 66: A well-described parent dataset from JECS.
- Line 66 – 67: N=80,000 was justified by estimation of <1% prevalence value. In fact, the cut off value was two standard deviations using a one sided Z distribution. Thus, the predetermined prevalence was about 2.3%, which was less than 2,000 cases per exposure category.
- Line 100 – 103: The “cleaning the bedroom floor with a vacuum cleaner” could be excluded using collinear analysis. The logic of running Spearman or Pearson to combine variables was understood.
Response: Thank you for these comments. - Line 110 – 116: “Culturally and developmentally appropriate stimulated activity” (Ref 15, p45) can be emphasized rather than arguing for “normal neurological development” (Line 114).
Response: We have removed “normal neurological development” from the sentence. On the other hand, “Culturally and developmentally appropriate stimulated activity” (Ref15, p. 45) might not be applicable to ASQ-3 because this description is about home visiting. Altogether, we have revised the sentence as follows: “The ASQ-3 is a set of well-validated, globally used questionnaires that is recommended by the United Nations Children’s Fund to identify potential delays and help determine which children need further assessment or ongoing monitoring [15].” (lines 111-114) - Line 125-126: Psychometrics are well-described.
- Line 131 – 157: A thorough explanation of covariates was appreciated.
Response: Thank you for these comments. - Line 160: Spell out “chi-square” before the Greek letter.
Response: We agree and added it. (line 160) - Results:
- Line 193: Please add an explanation that “an annual income” is for the household, per person, or for the mother.
Response: We have added “household” to the sentence. (line 195) - As for indentation, line 227 – 234 is a part of the caption for Figure 2.
- Two tables and one figure are adequate for the results. The supplemental tables are also well presented. The tree plot is very nice to have.
- Discussion:
- Line 259 – 265: the conservative cautionary summary statements were well-stated.
Response: Thank you for these evaluations. - Line 270 – 273: “In-depth studies” include, but are not limited to, sampling the infant and mother’s blood for known household neurotoxins.
Response: We have revised the sentence as follows: “Thus, further studies conducting randomized controlled trials to address causality, and in-depth studies, such as elucidating factors associated with reduced risk for developmental delay in relation to these interventions and sampling the infant and mother’s blood for known household neurotoxins, are warranted.” (lines 274-278) - Line 275 – 273: To clarify, the cleaning tasks do not have to be performed by the mother, but by any domestic partner/husband, care taker, home helper or older child.
Response: We have added the following sentence to the last part of the paragraph: “…, though the cleaning tasks do not have to be performed by the mother, but by any domestic partner/husband, caretaker, home helper, or older child.” (lines 287-289) - Line 303: “Other population” can refer to other cultural norms and languages used to describe the living environment. “Futon” can be a bed or sleeping mat, and “anti-mite covers” can be a bed-bug protector.
Response: We have revised the sentence as follows: “Finally, while our sample consisted of the JECS nationwide birth cohort, the extent to which our results can be generalized to other living environments, such as bed, sleeping mat, and bed-bug protector, cannot be determined.” (lines 308-311) - Conclusion: No comment
End of review.
According to your comments, our manuscript has improved considerably. Thank you for your very positive evaluation and valuable comments.
Please see the attachment.

Reviewer 2 Report
Congratulations on the authors' excellent work. Thank you for the opportunity to review this valuable addition to the literature regarding house dust avoidance during pregnancy and subsequent infant neuro-development. While the article is easy to read, well written, and follows a logical progression, there are some suggestions to improve readability and clarify information. I also recommended the authors use MeSH terms and improve searchability.
[abstract]
- Please identify the study design briefly in the abstract.
- Why did the authors choose to show the results of AORs in Table S5 rather than the numbers in Figure 2?
- Please, if possible, use MeSH terms as keywords to improve searchability.
[article]
- The sentence in lines 67-68 might be ambiguous. Please clarify and tell the reader more about the meaning of "with ≤ 1% prevalence."
- Please clarify that the cutoff point of the highest educational level "12" belongs to what category of the variable "highest education level." (line 141). It seems that "12" should not belong to level ≤12 (from line 192). Please also correct the category in Table 1.
- The area's development mentioned of the ASQ-3 should remain the same in line 117-120 and line 170-171 as the English version of ASQ-3, which means line 170: "general motor" should be changed to "gross motor" and "social-personal" should be changed to "person-social."
- Please clarify the covariate "pre-pregnancy body mass index." Because the author measured the participant's body mass index during the first trimester, is it suitable to call this covariate "PRE-pregnancy body mass index?" (line 133)
- Please identify the JPY means when it is first mentioned in this article (line 142.)
- Line 192 showed the percentage "35.0%" while Table 1 showed "35.1%". Please recheck the number.
- Please explain or show more details of lines 218-219. Where does the number "0.116±106" and "0.222±0.273" come from? What does the "case number" mean?
- Can the author briefly explain the differences between Figure 2 and Table S5?
- The authors have mentioned that "the adjusted ORs tended to decrease with an increase in cleaning frequency. Therefore, we found a negative dose-response relationship…." Could the author provide the trend test result for this dose-response effect? (line 251)
There were no apparent spelling or grammatical errors in this paper.
Author Response
Response to Reviewer 2 Comments
Comments and Suggestions for Authors
Congratulations on the authors' excellent work. Thank you for the opportunity to review this valuable addition to the literature regarding house dust avoidance during pregnancy and subsequent infant neuro-development. While the article is easy to read, well written, and follows a logical progression, there are some suggestions to improve readability and clarify information. I also recommended the authors use MeSH terms and improve searchability.
Response: Thank you very much for your high evaluation. Following your advice, we have selected keywords from MeSH terms.
[abstract]
- Please identify the study design briefly in the abstract.
Response: We have added the following sentence to the abstract. “Therefore, we examined this association by analyzing 81,106 mother-infant pairs who participated in a nationwide birth cohort in Japan.” (lines 19-20) - Why did the authors choose to show the results of AORs in Table S5 rather than the numbers in Figure 2?
Response: We were not aware of this issue. Surely, it is unnatural. Thus, we have revised this part as follows: “(AOR varied from 0.73 to 0.95, median: 0.84).” (line 25) - Please, if possible, use MeSH terms as keywords to improve searchability.
Response: As mentioned above, we have selected keywords from MeSH terms.
[article]
- The sentence in lines 67-68 might be ambiguous. Please clarify and tell the reader more about the meaning of "with ≤ 1% prevalence."
Response: Thank you for this indication. We have revised the sentence as follows: “The sample size was determined in advance to maintain adequate statistical power enough to detect rare diseases or conditions, such as with ≤ 1% prevalence.” (lines 65-66) - Please clarify that the cutoff point of the highest educational level "12" belongs to what category of the variable "highest education level." (line 141). It seems that "12" should not belong to level ≤12 (from line 192). Please also correct the category in Table 1.
Response: Thank you for this indication. The highest educational level "12" does not belong to “12–<16” but “≤12” category. Thus, we have replaced all the “12–<16” with “<12–<16” in the manuscript as well as in the supplementary tables. (line 141) - The area's development mentioned of the ASQ-3 should remain the same in line 117-120 and line 170-171 as the English version of ASQ-3, which means line 170: "general motor" should be changed to "gross motor" and "social-personal" should be changed to "person-social."
Response: Thank you for these indications. We have corrected them. (lines 172) - Please clarify the covariate "pre-pregnancy body mass index." Because the author measured the participant's body mass index during the first trimester, is it suitable to call this covariate "PRE-pregnancy body mass index?" (line 133)
Response: We were not aware of this inconsistency, but what we asked during the first trimester was pre-pregnancy BMI. Therefore, we have revised the relevant parts as follow: “…, pre-pregnancy body mass index asked during the first trimester (<18.5, 18.5–<25, or ≥25 kg/m2),…”. (line 133) - Please identify the JPY means when it is first mentioned in this article (line 142.)
Response: Thank you for this comment. Since JPY (Japanese Yen) appeared only once in the manuscript, we have defined it: (<4, 4–<6, or ≥6 million Japanese Yen). (lines 142-143) - Line 192 showed the percentage "35.0%" while Table 1 showed "35.1%". Please recheck the number.
Response: Thank you for this indication. We have checked it and changed “35.0%” to “35.1%”. (line 194) - Please explain or show more details of lines 218-219. Where does the number "0.116±106" and "0.222±0.273" come from? What does the "case number" mean?
Response: Following your indication, firstly, we have added this explanation to the “2.3. Statistical analysis” section as follows: “Outcome variables were the total number of screen-positive cases (ranging from 0 to 5), each at 6 and 12 months postpartum. For example, if an infant falls in cases of communication, fine motor, and problem solving, the total number equals 3.” (lines 169-170)
Furthermore, to explain "case number" in Figure 2 more appropriately, we have added it to the caption as follows: “Figure 2. Total number of screen-positive cases over the five developmental areas in the ASQ-3 (case number) and their odds ratios according to the exposure variables.” (lines 229-230) - Can the author briefly explain the differences between Figure 2 and Table S5?
Response: Figure 2 shows AORs for the total number of screen-positive cases over the five developmental areas in the ASQ-3. On the other hand, Table S5 shows AORs for screen-positive cases of each of the five developmental areas in the ASQ-3. We believe this question would already have been resolved by our response to comment #10 above. - The authors have mentioned that "the adjusted ORs tended to decrease with an increase in cleaning frequency. Therefore, we found a negative dose-response relationship…." Could the author provide the trend test result for this dose-response effect? (line 251)
Response: Thank you for this very important comment. We have added the results of trend test to the “3.2. Main analyses section,” where the results shown in Figure 2 are summarized (the last part of the second paragraph) as follows: “Trend test for adjusted models revealed that all p-values for trend were <0.001.” (lines 225-226)
Furthermore, we have added the results to Figure 2 and Table S5. The following sentence was also added to Figure 2 and Table S5: “P-values for trend were calculated for adjusted models.” (line 232)
There were no apparent spelling or grammatical errors in this paper.
According to your comments, our manuscript has improved considerably. Thank you for your very positive evaluation and valuable comments.
Please see the attachment.

Round 2
Reviewer 2 Report
Dear authors,
Congratulations for your excellent work.
There are two minor points you might need to recheck:
- Please REMOVE one "therefore" in LINE 19-20.
- Please recheck "the highest educational level in LINE 141 and Table 1." It might be >12-<16 which means more than 12 and less than 16, NOT "<"12-<16.
Author Response
Response to Reviewer 2 Comments
Comments and Suggestions for Authors
Dear authors,
Congratulations for your excellent work.
There are two minor points you might need to recheck:
- Please REMOVE one "therefore" in LINE 19-20.
- Please recheck "the highest educational level in LINE 141 and Table 1." It might be >12-<16 which means more than 12 and less than 16, NOT "<"12-<16.
Submission Date
09 March 2021
Date of this review
09 Apr 2021 16:43:25
Response: Thank you very much for your detailed check. I have corrected all the points, including sTables 1–4.
Please see the attachment.
